# Analyses of lncRNAs, circRNAs, and the Interactions between ncRNAs and mRNAs in Goat Submandibular Glands Reveal Their Potential Function in Immune Regulation

**DOI:** 10.3390/genes14010187

**Published:** 2023-01-10

**Authors:** Aili Wang, Jianmin Wang, Meina Mao, Xiaodong Zhao, Qing Li, Rong Xuan, Fajun Li, Tianle Chao

**Affiliations:** 1Shandong Provincial Key Laboratory of Animal Biotechnology and Disease Control and Prevention, College of Animal Science and Veterinary Medicine, Shandong Agricultural University, Taian 271000, China; 2Key Laboratory of Efficient Utilization of Non-Grain Feed Resources (Co-Construction by Ministry and Province), Ministry of Agriculture and Rural Affairs, Shandong Agricultural University, Taian 271000, China; 3Shandong Peninsula Engineering Research Center of Comprehensive Brine Utilization, Weifang University of Science and Technology, Shouguang 262700, China; 4Shandong Vocational Animal Science and Veterinary College, Weifang 261000, China

**Keywords:** lncRNA, circRNA, ceRNA network, immune related ncRNAs

## Abstract

As part of one of the main ruminants, goat salivary glands hardly secrete digestive enzymes, but play an important role in immunity. The immune function of goat salivary glands significantly changes with age, while the expression profile and specific function of non-coding RNA during this process are unknown. In this study, transcriptome sequencing was performed on submandibular gland (SMG) tissues of 1-month-old, 12-month-old, and 24-month-old goats, revealing the expression patterns of lncRNA and circRNA at different ages. A total of 369 lncRNAs and 1699 circRNAs were found to be differentially expressed. Functional enrichment analyses showed that the lncRNA regulated target mRNAs and circRNA host genes were significantly enriched in immune-related GO terms and pathways. CeRNA network analysis showed that the key differentially expressed circRNAs and lncRNAs mainly regulate the key immune-related genes *ITGB2*, *LCP2*, *PTPRC*, *SYK*, and *ZAP70* through competitive binding with miR-141-x, miR-29-y, and chi-miR-29b-3p, thereby affecting the natural killer cell-mediated cytotoxicity pathway, the T cell receptor signaling pathway, and other immune-related pathways. It should be noted that the expression of key circRNAs, lncRNAs, and key immune-related genes in goat SMGs decreased significantly with the growth of the goat. This is the first reporting of lncRNAs, circRNAs, and ceRNA network regulation in goat SMGs. Our study contributes to the knowledge of changes in the expression of non-coding RNAs during SMG development in goats and provides new insights into the relationship between non-coding RNAs and salivary gland immune function in goats.

## 1. Introduction

Non-coding RNAs (ncRNAs) play a very important role in life activities. They not only participate in development and differentiation, reproduction, and the process of cell apoptosis, but also have many biological functions, such as cell reprogramming. NcRNAs also participate in immune response and their expression levels are closely related to diseases, such as cardiovascular diseases, tumors, metabolic diseases, and infectious immune diseases, etc.

Long non-coding RNA (lncRNA) is a class of RNA that is longer than 200 nucleotides without coding protein ability due to the lack or absence of an open reading frame. The number of lncRNAs is far less than that of protein-coding genes, mRNA. In different cellular environments and biological processes, lncRNA is a key regulator of gene expression at both transcriptional and post-transcriptional levels, which is related to its diverse molecular mechanisms. LncRNAs act as miRNA sponges and inhibit the expression of miRNA, thereby regulating miRNA target mRNAs [1]. lncRNA lnc-DC (monocyte into dendritic cells identified, lnc-DC) expressed in human dendritic cells could bind to the STAT3 signaling molecule in the cytoplasm, which indicated that lncRNAs can directly interact with signaling molecules in the cytoplasm and up regulate their post-translational modifications to affect cell differentiation and function [2]. LncRNAs can regulate immune responses in infected animal models with the help of T cells [3].

Shen et al. analyzed the lncRNA expression profile of spontaneously hypertensive rats and normotensive rats by DNA microarray and bioinformatics methods, and 225 differentially expressed lncRNAs (DElncRNAs) were obtained. The potential functions of 10 DElncRNAs were closely related to the immune response process [4]. Fu et al. showed that lncRNA PVT1, which was upregulated in CD4 T cells from patients with Sjogren’s syndrome, can maintain the expression of *Myc*, and thus regulate CD4 T cell proliferation and effector function by regulating glycolytic reprogramming. Inhibition of glycolysis attenuates CD4 T cell proliferation and autoimmune responses in Sjogren’s syndrome [5]. In goats, DElncRNAs play important roles in growth regulation, cell morphological changes, and cell secretion [6]. LncRNA may be involved in the regulation of goat inner adipocyte maturation through growth and development, vesicle transport and fusion, and cell morphology [7,8]. LncRNA: XLOC_446331 regulated the development of adolescent goats, lnc_011371, lnc_007561, and lnc_001728 may play an important role in goat skeletal muscle [9], and ceRNAs (lncRNA-mRNA) may be involved in the lactation process [10]. Although a number of studies have shown that lncRNAs play an important role in regulating genomic activity, the expression of protein-coding genes, mRNA processing, and cell development, the exact functions and mechanisms of lncRNAs remain to be investigated.

CircRNA is a class of endogenous cyclic annular RNAs formed by reverse splicing. When circRNA was first discovered, it was considered a by-product of transcription [11], so it was classified as noncoding RNA (ncRNA, noncoding RNA). CircRNAs, widely exist in eukaryotes with a stable structure, are highly conserved, and have abundant expression [12]. In recent years, circRNA has become a research hotspot in the field of RNA [13], which has potential biological functions in animal reproduction. Germ cell growth and development were proved to be up-regulated by some circRNAs, while others down-regulate, and the mechanism was usually achieved by targeting miRNAs. CircRNAs have potential biological functions in animal reproduction [14]. By regulating the spongiform miRNAs in the circRNA-miRNA-mRNA co-expression network, circRNA achieves regulation of the expression of miRNAs and target genes [15]. The construction of the circRNA-miRNA-mRNA networks can better elucidate the role of circRNAs in various biological processes, which needed to be confirmed by more evidence. Wang et al. showed that circ-chr19 enhanced the expression of claudin-18 (*CLDN18*) by targeting miR-30b-3p, thus playing the role of ceRNA in Ebola virus infection [16]. Potential miRNA targets of hsa_circ_001937 participated in antimicrobial immune response by regulating the NF-κβ signaling pathway [17]. Spongy circRNAs in the circRNA-miRNA-mRNA co-expression networks can regulate the expression of miRNAs and target genes, indicating their potential role in transcriptional regulation [18]. The conservative, stable, and ceRNA characteristics of circRNA provide a new perspective and direction for studying the function of circRNAs.

Bioinformatics analysis is a way to better elucidate the interactions between different types of RNA. Transcriptional regulation is a complex network, and no type of RNA was isolated. It is not enough to analyze the biological mechanisms of the system just by studying one. Non-coding RNAs play a central role in regulating gene expression at multiple levels and can influence various aspects of cellular processes, including apoptosis, proliferation, cell cycle, migration, invasion, and drug resistance [19]. Non-coding RNAs participate in various biological processes and form complex regulatory networks through their interactions.

As the second largest glandular tissue in mammals, the SMG secretes saliva, immunoglobulin A, β defensin, and other biologically active substances [20,21]. In our previous studies, mRNAs and miRNAs were predicted to be involved in immune function [18,22]. In this study, the lncRNAs and circRNAs in different developmental stages of goat SMGs were studied, the immune-related circRNAs were screened, and the transcriptional regulation relationships among ncRNAs, core regulatory miRNAs, and mRNAs were predicted. The results of this study will enrich the expression differences and promote the understanding of goat SMGs at different developmental stages and provide reference information for molecular breeding and scientific prevention to improve goat immunity.

## 2. Materials and Methods

### 2.1. Ethics Statement, Library Construction and Alignment

As reported in our previous work [18], all animal experiments were approved by the Committee on the Management and Use of Laboratory Animals of Shandong Agricultural University (Permit Number: SDAUA-2018-052) and conducted in accordance with the ‘‘Guidelines for Experimental Animals’’ of the Ministry of Science and Technology (Beijing, China). Every effort was made to minimize the pain.

Sample collection, RNA extraction, library construction, alignment with the reference genome, and DElncRNAs and DEcircRNAs screening were implemented according to the method in our previous work [18]. To distinguish the samples, the SMGs samples from 1-month-old goats were labeled A1-L, A2-L, and A3-L; the samples from 12-month-old goats were labeled B3-L, B4-L, and B5-L; and the samples from 24-month-old goats were labeled C2-L, C3-L, and C5-L.

### 2.2. LncRNA and circRNA Prediction 

Two pieces of software, CNCI (version 2) (Utilizing sequence intrinsic composition to classify protein-coding and long non-coding transcripts) and CPC (CPC: assess the protein-coding potential of transcripts using sequence features and support vector machine) (http://cpc.cbi.pku.edu.cn/, accessed on 1 December 2020) were used to assess the protein-coding potential of novel transcripts with default parameters. In the meantime, novel transcripts were mapped to the SwissProt database to assess protein annotation. The intersection of both the non-protein-coding potential results and the non-protein annotation results was chosen as a lncRNA.

After the removal of low-quality reads, the high-quality pure reads were compared to the ribosome database by the software bowtie2, the reads with ribosomes were deleted, and the remaining data were compared to the goat reference genome (http://www.ncbi.nlm.nih.gov/genome/?term=goat, accessed on 20 December 2020) by the software TopHat 2.0.4 [23,24]. Then, we intercepted the ends of each unmapped reads (default: 20 bp), got anchors reads, and removed the sequence at both ends of those anchors reads. The reference genome was then compared again using bowtie, and the result was submitted to find anchors_reads_circ for the identification of circRNAs. The above results were filtered with the conditions: breakpoints = 1, anchor_overlap <= 2, edit <= 2, n_uniq > 2, bes8t_qual_A > 35 or best_qual_B > 35, n_uniq > int (samples/2), and the length of circRNAs were shorter than 100 k; then the results were calculated.

### 2.3. Screening of DElncRNAs and DEcircRNAs

The lncRNAs with medium and high expression (Fragments Per Kilobase per Million, FPKM ≥ 50) were counted. EdgeR was used to analyze the expression differences between groups. DElncRNAs were screened under the conditions of FDR < 0.05 and |log2FC| > 1. The transcript expression level of each sample was log2, and hierarchical clustering analysis was performed on the samples and transcripts. The expression quantity (FPKM) of the 9 groups were Z-Score standardized by the R-language Scale function, and the heat map was drawn by the pheatmap package. 

To circRNA, the expression level was calculated and normalized by back-spliced Reads Per Million (RPM). DEcircRNAs were screened under the conditions of *p* < 0.05 and |log2FC| > 1.

### 2.4. lncRNA and circRNA Family Analysis 

The software Infernal (http://eddylab.org/infernal/, accessed on 10 January 2021) was used in sequence alignment to better annotate lncRNAs at the evolution level (Infernal 1.1: 100-fold faster RNA homology searches), and lncRNAs were classified by secondary structures and sequence conservation. CircRNA types, including annot_exon, one_exon, exon_intron, intronic, antisense, and intergenic, were identified by combining the circRNA information given by find_circ with the transcriptome annotation of the reference genome.

### 2.5. Association Analysis of lncRNA and mRNA

The interaction modes of lncRNA and mRNA can be divided into antisense, cis-regulation, and trans-regulation. In order to identify them, functional enrichment analysis was performed on different target mRNAs.

Analysis of antisense lncRNAs; to reveal the interaction between antisense lncRNAs and mRNAs, we used the software RNAplex to predict the complementary base binding relationship between antisense lncRNAs and mRNAs. The ViennaRNA package was used to calculate the minimum free energy from its thermodynamic structure to predict the best base pairing. 

LncRNA cis-regulation analysis; the lncRNAs that were annotated as “unknown regions” in the previous analysis were annotated. If they are located within 30 kb upstream or downstream of a gene, these lncRNAs are likely to intersect with cis-acting elements and thus participate in transcriptional regulation.

LncRNA trans-interaction analysis; the prediction of target gene pairs was performed by correlation analysis or co-expression analysis of lncRNA and protein coding genes between samples. The group size of this study was 9, so the Pearson correlation coefficient method (represented by r) was used to analyze the correlation between the expression of lncRNAs and mRNAs among groups. To predict the main function of lncRNAs, the mRNAs with an absolute value of correlation greater than 0.999 were selected.

### 2.6. GO and KEGG Pathway Enrichment Analyses 

Referring to our previous methods [18], the GO (Gene Ontology) and KEGG (Kyoto Encyclopedia of Genes and Genomes) pathway enrichment were performed on the predicted antisense binding target mRNAs, cis-acting target mRNAs, selected mRNAs with r > 0.999 in the trans-interaction analyses, and host genes for DEcircRNAs with FDR ≤ 0.05.

### 2.7. Screening of Immune-Related circRNAs

Immune-related circRNAs were screened from the significantly enriched pathways, and the network was constructed by Cytoscape 3.7.2 software. 

### 2.8. Validation of circRNAs 

Six DEcircRNAs, novel_circ_012271, novel_circ_002834, novel_circ_002839, novel_circ_006927, novel_circ_003764, and novel_circ_009665 were randomly screened from the transcriptome sequencing results for circle structure validation and RT-qPCR analysis. The primers used for RT-qPCR analysis were designed according to the sequences of selected DEcircRNAs using Primer Premier 5.0 software, which were listed in Table 1. Total RNA was extracted from the 9 samples using TRIzol reagent (Invitrogen, Carlsbad, CA, USA), then divided into two parts, one was treated with Rnase R. A reaction volume of 20 µL with 2 µL Rnase R(20 U/μL), 2 µL 10× Reaction buffer, 5 μg RNA, and ddH_2_O was used. Another part was stored in −80 °C. cDNA, Dalian, was synthesized by using the Primerscript RT reagent Kit (TaKaRa, Dalian, China) with 1 µg of total RNA, then the circRNA amplification were performed by the primers, and the circle structures were detected by Nucleic Acid electrophoresis. RT-qPCR was performed using the SYBR Premix Ex Taq II (TaKaRa, DRR081A) and conducted using a Stratagene Mx3000 real-time system (USA). A reaction volume of 20 µL with 1 µL forwarding primer (0.4 mmol each), 1 µL reverse primer (0.4 mmol each), 10 µL 2× Es Taq MasterMix, and 6 µL RNasE-free ddH_2_O was used. The reaction mixtures were incubated in a 96-well plate at 94 °C for 5 min, followed by 30 cycles of 94 °C for 30 s, 57 °C for 30 s, and 72 °C 3 min. All reactions were performed in triplicate with GAPDH used as the reference gene. The 2^−ΔΔCT^ method was used to calculate the results according to our previous study (peer J). Significance of gene expression between different groups was tested by Tukey’s HSD and gene expression histograms were performed by GraphPad Prism 8 software.

### 2.9. Analysis of ceRNA Regulatory Networks

Mireap (version 0.20), miRanda (version 0.10.80), and TargetScan (version 8.0) were used to perform target gene prediction, and the intersected prediction from the three software was used as the final target gene prediction of miRNAs. The potential ceRNAs were screened using the following methods. Including the targeting interaction and expression levels of miRNAs and candidate ceRNAs were negatively correlated, there was a positive correlation between the expression levels of candidate ceRNAs and the degree of enrichment of candidate ceRNAs binding to the same miRNAs. Based on the results, the ceRNA pairs with *p* < 0.05 were screened as the final ceRNA pairs using the hypergeometric test. The interactions between the previously screened core mRNAs and ncRNAs were focused and screened, and cytoscape 3.7.2 was used to construct a ceRNA regulatory network.

### 2.10. circRNA-miRNA-mRNA Network Analysis

Immune-related DEcircRNAs in significantly enriched pathways were screened. Targeting relationships among immune-related DEcircRNAs, DEmRNAs, and DEmiRNAs were predicted by RNAhybrid (version 2.1), miranda (version 0.10.80), and targetscan (version 8.0) software. The result obtained from the intersection was predicted as a targeted relationship, and the network was constructed by the cytoscape 3.7.2 software. 

## 3. Results

### 3.1. Sequencing Data Statistics

A total of 3192, 2936, and 2730 lncRNA transcripts were identified, including 1627, 1492, and 1381 known lncRNA transcripts in groups A, B, and C, respectively. They accounted for 60.80%, 55.75%, and 51.61% of the total lncRNA transcripts, respectively. The number of new lncRNA transcripts in the three groups was 1565, 1444, and 1349, as shown in Appendix A. A total of 1728 new lncRNAs were identified (Figure 1a). In all 9 samples, the proportion of known lncRNA transcripts were all less than 50%, as shown in Appendix A.

A total of 17,263 circRNAs were identified in 9 samples. Since goat circRNAs were not included in the circBase or PlantcircBase databases, the identified circRNAs were all defined as new circRNAs.

### 3.2. Novel Transcript Identification and Type Statistics

According to their location relative to protein-coding genes, lncRNAs were classified into five classes: intergenic lncRNAs, bidirectional lncRNAs, intronic lncRNAs, antisense lncRNAs, and sense overlapping lncRNAs. The new lncRNA transcript types identified in the 9 libraries were statistically analyzed, as shown in Figure 1b. The largest number of new lncRNAs was Intergenic lncRNAs, 1055, followed by Bidirectional lncRNAs, with a number of 261. In addition, the numbers of antisense lncRNAs, sense overlapping lncRNAs, and other types of transcripts were 202, 149, and 61, respectively. No intronic lncRNAs were found in goat submandibular glands.

As shown in Figure 1c and Appendix A, the trend of lncRNA length change in all lncRNAs, DE lncRNAs, and novel lncRNAs were basically the same. Among all lncRNAs, the number of lncRNAs with 200–1200 nt length accounted for 50.98% of the total lncRNA number, and the number of lncRNAs with 301–400 nt length was 274, which was the largest. Among the novel lncRNAs, the sum of lncRNAs with the lengths of 300–1300 nt and 1600–1900 nt was 849, and this was the highest, accounting for 49.13% of the total number of novel lncRNAs. Among the DElncRNAs, there were 189 DElncRNAs with lengths of 400–900 nt, 1100–1900 nt, and 2200–2500 nt, accounting for 51.36% of the total DElncRNAs. We predicted that DElncRNAs with different length ranges may act differently with mRNAs and then have different functions.

As shown in Figure 2a, the circRNA host genes were distributed on 29 chromosomes and 1127 circRNA host genes were located on chr 1, which was the most numerous circRNA source chromosome, indicating that the main source of goat SMG circRNAs was chr 1. The main circRNA type was annot_exons (Figure 2b), with a statistical number of 11,663, followed by exon_intron, with a number of 1415, and the numbers of antisense, intergenic, one_exon, and intronic were 1391, 1036, 921, and 840, respectively. As Figure 2c showed, the overall changes in the distribution of the number of circRNAs on the same chromosome were consistent as A > B > C, indicating that the number of circRNAs in the SMGs of goats gradually decreased with age. The number of circRNAs in the range of 300–400 nt was the highest, accounting for 12.18% of total, followed by circRNAs in the range of 400–500 nt, accounting for 10.88% (Figure 2d). The number of circRNAs in the length of 100–600 nt goat submandibular gland accounted for more than 40% of all.

### 3.3. DElncRNAs and DEcircRNAs Analysis

A total of 369 DElncRNAs were identified between groups, and the results are shown in Table 2 and Figure 3a,c–e. Among them, 288 were between groups A and B. There were 262 between groups A and C, only 1 between groups B and C, and they were highly expressed in group B (Figure 3b). Cluster analysis showed that all the DElncRNAs clustered well, indicating that the lncRNAs expressed in goat SMGs were stable at all three developmental stages, since the differences between groups were large, Figure 3f. Among them, the expression patterns of lncRNAs in groups B and C were highly similar, but very different from those in group A. This suggested that the lncRNA expression pattern changed dramatically in goat SMGs with growth. 

There were 833, 780, and 86 DEcircRNAs screened between groups A and B, group A and C, and group B and C, respectively. The results are shown in Figure 4 and Table 3. Among all the significantly differentially expressed circRNAs, the top 10 with the most significant differences were significant DEcircRNAs. Type statistics were performed on DEcircRNAs among the groups (Figure 4e). Among groups A and B, and A and C, the main type of DEcircRNAs were annot_exon, followed by intergenic.

### 3.4. lncRNA-mRNA Association Analysis

Different interaction modes of all DElncRNAs-mRNAs were predicted and summarized, and the results are shown in Table 4. Antisense lncRNAs bind to sense mRNAs to regulate gene silencing, transcription, and mRNA stability. In this study, we predicted that 11 DElncRNAs regulate 10 mRNAs by antisense binding, 51 DElncRNAs regulate 70 mRNAs by cis-regulation, and 160 DElncRNAs regulate 141 mRNAs by trans-regulation. To further analyze the functions of these target mRNAs, GO and KEGG pathway enrichment analyses were performed with different interaction modes, respectively.

### 3.5. Enrichment Analysis of Antisense lncRNA Target mRNAs

GO and KEGG pathway enrichment analyses of antisense lncRNA target mRNAs help us understand the biological functions of DElncRNAs. As shown in Figure 5, 11 antisense lncRNA target mRNAs were annotated into 33 significantly enriched GO terms (*p* < 0.05), including 30 biological process GO terms and 3 molecular function GO terms (Figure 5a). The top four most significantly enriched GO terms were related to protein regulation and transport, which indicated that the functions of these 11 antisense lncRNA target mRNAs are protein transport and localization in biological processes. Among them, XR_310691.3, XR_001919471.1, and XR_001919469.1 targeted *SMPD3*, which was significantly enriched in the lipid metabolism pathway, lipid activity, and protein metabolism (Table 5). The results indicated that the 3 DElncRNAs took part in regulating lipid digestion.

KEGG pathway enrichment analysis showed that a total of 3 pathways were significantly enriched (*p* < 0.05), as shown in Figure 5b. TCONS_00044048 antisense binding target gene ncbi_100750237 (symbol: TG, thyroglobulin) significantly enriched thyroid synthesis and autoimmune thyroid disease pathways, indicating that TCONS_00044048 plays an important role in thyroid disease.

### 3.6. Enrichment Analysis of lncRNA Cis-Regulated Target mRNAs

Enrichment analysis of 73 predicted DElncRNAs cis-acting target mRNAs was performed (Figure 5c). It’s worth noting that all the significantly enriched biological process GO terms were protease activation terms, and the most significantly enriched term was protein tyrosine kinase activity (*p* = 6.00 × 10^-4^). The XM_018060897.1 cis-acting target gene ncbi_108637839 (symbol: *MIF*) was significantly enriched in terms of positive regulation of fatty acid transport and positive regulation of lipid transport.

KEGG pathway enrichment analysis showed that lncRNA cis-acting target mRNAs were significantly enriched in 17 pathways (*p* < 0.05) (Figure 5d). The top four pathways with the highest significance were NF-κβ signaling pathway, T cell receptor signaling pathway, intestinal immune network producing IgA, and antigen cell processing and presentation, all of which were immune-related pathways. In order to further study their functions, lncRNAs in the first four significantly enriched pathways were analyzed, and the results are shown in Table 6. Among them, only XR_001917128.1 was known lncRNA. The lncRNAs (r > 0.9, *p* < 0.05) that were highly significantly associated with cis-regulated target mRNAs are italicized, and they were TCONS_00028592, TCONS_00013148, and TCONS_00029100, respectively.

### 3.7. Enrichment Analysis of lncRNA Trans-Interacted Target mRNAs

Enrichment analysis was performed on 161 lncRNAs and their trans-interacted mRNAs. There were 9 significantly enriched pathways (*p* < 0.05) (Figure 5f), including 6 immune system pathways. The three most significantly enriched pathways were Antigen processing and presentation, Cytokine-cytokine receptor interaction, and Hematopoietic cell lineage. By analyzing the immune-related pathways, 13 immune-related lncRNAs that trans-interacted with mRNAs were predicted, including 10 new lncRNAs. All the Pearson correlation coefficients between lncRNAs and positive trans-interacted target mRNAs were higher than 0.99, while the Pearson correlation coefficient between lncRNAs and reverse co-expression mRNAs was lower than −0.96 (Table 7). GO enrichment analysis is shown in Figure 5e. There were 146 significantly enriched terms, including 28 molecular function GO terms, 16 cell component GO terms, and 102 biological process GO terms. The trans-interacting target mRNAs of 29 lncRNAs were significantly enriched in three biological metabolism processes: the organic matter metabolism process, the macromolecular matter metabolism process, and the protein metabolism process, indicating that these lncRNAs were related to macromolecular matter, organic compounds, and protein metabolism.

### 3.8. Enrichment Analysis of DEcircRNA Host Genes

To understand the functions of DEcircRNAs, host genes were predicted and GO and KEGG enrichment analyses of host genes were performed, as shown in Figure 6. The DEcircRNA host genes in A vs. B were significantly enriched in 42 GO terms (FDR < 0.05). It is worth noting that they were significantly enriched in 3 terms: regulation of the metabolic process, regulation of the cellular metabolic process, and regulation of the macromolecule metabolic process, which may be related to digestion and metabolism of nutrients. As shown in Figure 6b–d, 41, 25, and 1 pathways of DEcircRNA host genes were significantly enriched between groups A and B, A and C, and B and C, respectively. Among the significantly enriched pathways, four immune-related pathways were included, they were: the B cell receptor signaling pathway, leukocyte transendothelial migration, natural killer cell mediated cytotoxicity, and platelet activation.

### 3.9. Screen of Immune-Related circRNAs

To further study the function of the host genes, which were significantly enriched in immune-related pathways, the host gene-pathway-circRNA interaction network was constructed (Figure 7a), which contained 39 DEcircRNAs, 39 host genes, 7 immune-related significantly enriched pathways, and 206 interactions. The top 10 immune-related DEcircRNAs with the highest expression levels involved in the network were summarized and statistically analyzed, as shown in Table 8; they were most significantly enriched in the Leukocyte transendothelial migration KEGG pathway and immune response-regulating cell surface receptor signaling pathway GO terms.

### 3.10. Validation of circRNAs 

As shown in Figure 8, the six circRNAs, novel_circ_012271, novel_circ_002834, novel_circ_002839, novel_circ_006927, novel_circ_003764, and novel_circ_009665, were ring structures and the qRT-PCR verification results of circRNA (Figure 9) were basically consistent with the RNA-seq sequencing results.

### 3.11. Analysis of ceRNA Regulatory Networks

LncRNA-miRNA-mRNA and circRNA-miRNA-mRNA network predictions were performed on all differentially expressed RNAs, and the results showed that the interaction numbers were large (Appendix A). It was complicated if all the interactions were used to build networks, so we focused on the interactions between core mRNAs and ncRNAs obtained from previous analyses. After targeting relationship predictions, 81 miRNAs were targeted to regulate 9 core mRNAs, among which 9 miRNAs reached significant negative correlation with core mRNAs (*p* < 0.05), and Pearson correlation coefficients were all less than −0.63. Among them, miR-141-x targeted to *ITGB2*, *LCP2*, *PTPRC*, *SYK*, and *ZAP70*,miR-29-y targeted to *PTPRC* and *ZAP70*, and chi-miR-29b-3p targeted to *PTPRC* and *ZAP70*, where they all reached high correlation (r < −0.9, *p* < 0.001) (Table 9), and miR-141-x and miR-29-y were all core miRNAs. To further study the regulatory relationships of ceRNAs with miR-141-x, miR-29-y, and chi-miR-29b-3p we combined the circRNA database results for the prediction of targeting relationships and Pearson correlation coefficient calculation. For miR-141-x, miR-29-y, and chi-miR-29b-3p, 47, 28, and 57 significantly regulated circRNAs with r < −0.65 were predicted (*p* < 0.05), respectively, and among them, 6, 4, and 5 with r < −0.9, respectively (Appendix A). A total of 32, 31, and 33 (*p* < 0.05) significantly regulated lncRNAs were screened for miR-141-x, miR-29-y, and chi-miR-29b-3p, respectively, (r < −0.65), and among them 6, 10, and 8 with r < −0.9, respectively (Appendix A). With a Pearson correlation r value for miRNA expression of less than −0.9, 13 key circRNAs and 14 key lncRNAs were identified, and their regulatory relationship is shown in Figure 7b.

### 3.12. circRNA-miRNA-mRNA Network Anaylysis

To study the regulatory relationships between immune-related circRNAs, core miRNAs [22], and DE mRNAs, the targeting relationship between them was screened with r < −0.95 and PDF < 0.05. As shown in Figure 7c, the miRNA with the highest quantity of targeted mRNAs was miR-29-y, suggesting that miR-29-y plays a key role in gene regulation, but the specific regulatory mechanism needs further study.

## 4. Discussion

### 4.1. lncRNAs Were Invoved in Immune-Ralated Pathways

Long non-coding RNAs, defined as transcripts with a length greater than 200 nucleotides and a lack of protein-coding ability, are a group of non-coding RNAs with the largest number of bases generated in the genome. LncRNAs regulate gene expression at transcriptional and post-transcriptional levels through a variety of mechanisms. They play a key role in biological processes and diseases such as cell differentiation, tissue and organ development, and cancer metastasis [25]. A number of lncRNAs have been identified from humans, rats, mice, bovines, sheep, goats, and other species by high-throughput sequencing [8]. For goats, data have also been reported on lncRNAs in skeletal muscle [26], hair follicles [27], mammary glands [28], and other tissues, but no reports have been made on lncRNAs in the submandibular glands. Although the role of coding genes in immune cell function has been well described, the study of immune-related lncRNAs has just begun to emerge [29]. LncRNAs are a key regulator of gene transcription during inflammatory responses. In the innate immune system, lncRNAs play a role by regulating the inflammatory process [30]. In the acquired immune system, the function of *NFAT1* in T cells was regulated by lncRNA: NRON (ncRNA repressor of NFAT) [31]. Maladjusted lncRNAs in human diseases such as inflammatory bowel disease, diabetes, allergies, asthma, and cancer have been found to be crucial for immune mechanisms. They are involved in cell differentiation, migration, and production of inflammatory mediators by regulating protein-protein interactions or by their assembly with RNA and DNA, and finally, they may affect all possible lncRNA biological effects through cis or trans interactions [32].

In this study, a total of 4404 lncRNAs were identified from the SMGs of 1-month-old goat kids, 12-month-old unbred adolescent goats, and 24-month-old adult goats (three samples in each group) by high-throughput sequencing. The number of known lncRNA transcripts in the three groups accounted for 60.80%, 55.75%, and 51.61% of the total lncRNA transcripts, respectively. This indicates that the number of known lncRNAs obtained by goat SMGs accounts for more than half of the total number of lncRNAs. There were only 11 highly expressed lncRNAs, indicating that the expression of lncRNAs was generally low in different developmental stages of goat SMGs. The results of differential expression analysis indicated that the expression pattern of lncRNAs in the goat SMGs was similar between adolescent and adult goats, but the expression pattern of lncRNAs changed greatly from 1-month-old goat kids to adolescent goats, which may indicate that the expression of lncRNAs is time and tissue specific during the growth and development of goat SMGs.

To predict the function of lncRNAs, predictive analysis of the interaction mode of all DElncRNAs and enrichment analysis for targeted mRNAs with different modes were performed. The results showed that the targeted mRNAs with different modes were partially enriched in protein metabolism, lipid metabolism, and other related pathways, indicating that the expression of digestive enzymes in goat SMGs was regulated by lncRNAs. Significant enrichment results showed that the main mode of lncRNA regulation of digestive enzymes was cis-acting.

The top three pathways significantly enriched for cis-acting target genes, the NF-κβ signaling pathway, the T cell receptor signaling pathway, and the intestinal immune network producing IgA, are all immune-related pathways, and their target genes are all up-regulated, indicating that the lncRNAs are involved in the three pathways in SMGs by up-regulating target mRNAs through cis-acting mode. We noticed that about one-third of the functions of trans-acting lncRNA targeted mRNAs were immune-related biological processes, and most of the significantly enriched pathways were also immune-related pathways. Therefore, we predict that DElncRNAs in goat SMGs have a regulatory role in immune correlation, but the specific regulatory mechanism needs to be further studied. As core genes in the NF-κβ signaling pathway and the T cell receptor signaling pathway, *V-TCR* were cis-regulated by TCONS_00013148 and trans-regulated by TCONS_00041162 and TCONS_00052910. As well as *TCRB,* a core gene in the above two pathways, was trans-regulated by XR_001297267.2. TCR (T-cell antigen receptor) is a heterodimeric glycoprotein composed of two peptide chains, TCR-α and TCR-β. Each peptide chain is generated by genomic rearrangements of the variable, diverse, connecting, and constant regions segments, and TCR-β contains 55 different V regions [33]. The results suggested that different types of lncRNAs may regulate different V regions of the TCR-β chain in the process of regulating *V-TCR.* Fyn is a member of the Src family tyrosine kinases with a molecular weight of 59 kDa and has diverse biological functions [34]. The study by Oers et al. implied a critical role for Lck and Fyn in T-cell development [35]. The TCONS_00072725 cis-acting target gene *PKCƟ* (encoded by PRKCQ) played a key role in regulating the differentiation and proliferation of T lymphocytes [36], and its mutation can cause the occurrence of Crohn’s disease and other diseases [37]. Prediction results have showed that TCONS_00029100 further affected the function of NKG2D through cis-binding to the target gene NKG2D ligand, LOC102185264. It can be seen that TCONS_00013148, TCONS_00041162, TCONS_00052910, TCONS_00072725, XR_001297267.2, and TCONS_00029100 cis/trans target mRNAs were related to the immune function of T cells. Moreover, these 6 lncRNAs were highly expressed in the SMGs of 1-month-old goat kids and lowly expressed in 12-month-old unbred adolescent goats and 24-month-old adult goats, which may be one of the reasons that the expression of immune-related genes was significantly down-regulated with the growth and development of goat SMGs.

Membrane-spanning 4-domain family gene (*MS4A*) is mainly expressed in lymphoid nuclei and blood, and plays a key role in regulating cell activation, growth, and development. Unlike other lncRNAs, TCONS_00078593, which trans-acted with target gene *HSP90AA1* (which encode the heat shock protein Hsp90α), was significantly down-regulated in the SMGs of 1-month-old goat kids and up-regulated in 12-month-old unbred adolescent goats and 24-month-old adult goats. Previous studies have found that Hsp90α is associated with scrapie in goats, thus, TCONS_00078593 might be a biomarker for scrapie in goats.

Chemokines are essential for the tissue-specific migration and localization of immune cells in lymphoid organs. Without inflammation, the receptor CCR4 is predominantly expressed by regulatory T cells (T Reg) [38]. By researching the CCL22-deficient mice, Moritz Rapp et al. found that the expression of *CCL22* by dendritic cells (DCS) promoted cell-to-cell contact and interaction with regulatory T cells (T Reg) through the CCR4 receptor, and CCL22-deficient mice had increased susceptibility to inflammatory diseases. Therefore, the CCL22-CCR4 axis is an important immune checkpoint to control T-cell immunity [39]. Tyner et al. found that when mice lacking the chemokine CCL5 were infected by parainfluenza or human influenza virus, impaired immunity leads to viral clearance delaying, excessive airway inflammation, and even respiratory death [40]. In this study, we found that XR_310184.3 and TCONS_00072236 trans-acted target genes *CCL22* and *CCR4*. Therefore, the co-expression of these two lncRNAs may affect the immune function of T cells. Moreover, we found they were significantly down-regulated in 12-month-old unbred adolescent goats and 24-month-old adult goats, which may be another influencing factor for the down-regulation of immune-related genes in goat SMGs. Chemokine CCL5, after binding with receptor CCR5, can activate the PI3K-AKT and MEK-ERK signaling pathways to play an immune protective role. However, in goat SMGs, we found that the significantly upregulated TCONS_00020715 trans targeted gene *CCL5* was negatively correlated, which seems to indicate that the ability of goat SMGs to initiate related signaling pathways through CCL5 to cope with viral infection is weakened with age. Interleukin 7 (IL-7) and its receptor are formed by IL-7Rα (encoded by *IL-7R*) and γc, which are essential for normal T cell development and homeostasis [41]. TCONS_00035813, which regulated *IL-7R*, was similar to other lncRNAs that affect T cell function, and its expression was gradually down-regulated with the increase in age. In conclusion, all lncRNAs related to T cell development, differentiation, and immune function in goat SMGs down-regulate the expression of target genes, which may be a major reason for the down-regulation of immune-related genes during goat submandibular gland development. 

Among all lncRNA target genes that were significantly enriched in immune pathways, *MS4A1* was a gene related to B cell immune function. MS4A1 (CD20) is selectively expressed on mature B cells and most malignant B cells, and has become a clinical target for the treatment of B-cell lymphoma and some autoimmune diseases [42]. The trans-acting lncRNA XR_001919671.1 of *MS4A1* was gradually down-regulated with the growth and development of goat SMGs, which reduced the risk of autoimmune diseases in goat SMGs to some extent. Therefore, further studies on the regulatory principle of XR_001919673-1 in the treatment of B-cell lymphoma and some autoimmune diseases may provide new insights for the treatment of B-cell lymphoma and some autoimmune diseases.

By high-throughput sequencing, we identified 17 immune-related lncRNAs by transcriptome analysis of SMGs in 1-month-old kids, 12-month-old unbred adolescent goats, and 24-month-old adult goats. The lncRNAs were all related to the immune function of T cells or B cells. The results predicted the possible changes of immune-related genes in goat SMGs with growth and development from the perspective of lncRNAs. Some of these target genes are closely related to the occurrence and development of diseases. Through further verification, these lncRNAs may become biomarkers for improving autoimmunity and disease treatment.

### 4.2. circRNAs Were Invoved in Immune-Ralated Pathways

With the development of RNA-sequencing and computer technology, the study of circRNAs is gradually becoming a hot topic. A variety of mammalian-derived circRNAs, including human [43], goat [9], mouse [44] and pig [45], have been identified. CircRNAs have also been identified in goat hair follicles [46], mammary glands [47], skeletal muscle [9] and endometrium [48], while studies on goat submandibular gland circRNAs have not been reported. CircRNA expression had high spatiotemporal specificity, and it is crucial to study these molecules at different development stages in different tissues. In this study, a total of 17,263 circRNAs were identified from nine libraries in three developmental stages of the goat SMGs by high-throughput sequencing. Analysis of the chromosomal origin of circRNAs revealed that all 29 autosomes of goats could produce circRNAs, with chromosome 1 producing the largest number of circRNAs, and chromosome 1 was previously found to be the main source of circRNAs in goat skeletal muscle [9], suggesting that chromosome 1 in goats was closely related to the production of circRNAs. The length of circRNAs was mainly concentrated in the range of 100–600 nt, with the largest number between 300–400 nt. Among all circRNAs, annot_exons type circRNAs were the most numerous, and a similar phenomenon was observed in DEcircRNAs. We predicted that these types of circRNA-originating genes have a higher coding capacity and deserve our attention.

The result of DEcircRNA analyses showed that the expression pattern of submandibular gland circRNAs in adolescent and adult goats was similar, while it was quite different from that of goat kids. The enrichment analysis of DEcircRNA host genes showed that all the significantly enriched terms were biological processes, with all three of the top terms being cancer-related pathways, and all the circRNA host genes were upregulated. The phosphoinositide 3-kinase (PI3K) has the function of catalyzing the production of phosphatidylinositol-3,4,5-trisphosphate in cell survival pathways, regulating gene expression and cellular metabolism, cytoskeletal rearrangement, and binding to catalytic subunits p110α, β, δ. and γ [49]. In this study, differentially expressed novel_circ_000481 host gene *PIK3CA* (encoding p110) and novel_circ_013299 host gene *PIK3R1* (encoding p85α) were significantly enriched in the B-cell receptor signaling pathway, glioma, the FoxO signaling pathway, prostate cancer, and other pathways, which were up-regulated in adolescent goats. Engelman showed that T antigens in polyomaviruses require physical interaction with PI3K to transform cells [50]. Overactivation of p110δ also caused T cell senescence, lymphadenopathy, and immunodeficiency [51]. Some cancers have activating mutations in the PI3K regulatory subunit P85α (encoded by *PIK3R1*). In wild-type PI3K holoenzymes, p85 inhibited p110α through intermolecular interactions, and this inhibition was relieved by a conformational change caused by the binding of the p85 amino-terminal SH2 structural domain to phosphotyrosine [52]. Therefore, we speculate that novel_circ_000481 and novel_circ_013299 may be closely related to immune regulation in goats, and their low expression in theSMGs of goat kids may be a protection.

It’s worth noting that DEcircRNA host genes were significantly enriched in immune-related pathways. To further study the immune-related circRNAs, the top 10 DEcircRNAs for all significantly enriched immune-related pathways were screened as core immune circRNAs based on the expression level, and their target genes were *NFATC3*, *LOC102189946, CR2*, *TXK*, *LOC102180664*, *PLCG2*, *ARHGAP5*, *ARHGAP35*, *PIGR*, and *LOC100860813,* respectively. Complement receptor type 2 (CR2; CD21), with an immunomodulatory effect, was found on normal T lymphocytes, which may play a role in regulating the immune response function of T cells and cellular susceptibility to lymphophilic viral infection [53]. CD19 is a B cell-restricted membrane protein in the immunoglobulin superfamily that is associated with the antigen receptor complex. CR2 (complement receptor type 2, CD21) allows the non-immune attachment of CD19, which may facilitate the interaction of B cells with other essential cells for cell activation [54]. Txk is an important regulator of cytokines produced by the CD4+ effector T cell population [55]. Phospholipase Cγ2 (PLCγ2), encoded by PLCG2, is associated with human urticaria, immunodeficiency, and autoimmune diseases. Microglia-mediated innate immune responses, which were led by PLCG2 mutations, promote the development of Alzheimer’s disease [56]. *CR2*, *NFATC3*, *TXK*, and *PLCG2* were regulated by novel_circ_011088, novel_circ_ 012271, novel_circ_004727, and novel_circ_012149, respectively. The four circRNAs were all highly expressed in goat kids and lowly expressed in adolescent and adult goats, which may indicate that the expression of immune-related genes in the SMGs of goats was down-regulated with age, and this was consistent with the analysis of DE mRNAs in goats.

P190RhoGAP-A (glucocorticoid receptor DNA-binding factor 1, or GRLF1, p190A) and p190RhoGAP-B (p190B), which were two isoforms of the p190RhoGAP protein, were encoded by *ARHGAP35* and *ARHGAP5*, respectively, and p190RhoGAP was considered as a major negative regulator of RhoA. Through its role in cytoskeleton remodeling, P190RhoGAP may be related to the pathogenesis of other stress-related and neurodegenerative diseases [57]. *ARHGAP35* has also been proposed as an important new cancer gene [58]. Similarly to P190A, the overexpression of p190B was associated with the expression of *MCT1* (multiple copies in T-cell malignancy-1, MCT1) in breast cancer [59]. In vitro MCT1-Src-p190B interaction caused tumoral multinucleated formation in breast cancer cells [60], which has also been demonstrated in nasopharyngeal and lung cancer. The high expression of *ARHGAP35* and *ARHGAP5* in the SMGs of adolescent goats suggested that the pathogenic genes in adolescent goats were different from those in goat kids and adult goats.

In organisms, secretory IgA (SIgA) antibodies represent the first line of antigen-specific immune defense. Most pathogens establish infection through mucous membranes, while secretory IgA (SIgA) antibodies play an “immune rejection” role in humoral defense [61] to protect the mucosal surface from environmental pathogens and antigens, and to maintain the homeostasis of symbiotic microorganisms. The previous study results of the goat SMG transcriptome in different developmental stages showed that the expression of IgA in adolescent goats was significantly higher than that of goat kids. It was the same with novel_circ_011081, which derives from the host gene pIgR. Therefore, we predict that novel_circ_011081 may play a certain role in preventing oral infectious diseases in adolescent goats.

By transcriptome analysis, 10 immune-related circRNAs were identified from goat submandibular glands; this was the first time to analyze the immune-related circRNAs in the SMGs of goats. These data provided new insights into the function of circRNAs in the submandibular gland of goats at different developmental stages.

### 4.3. CeRNA Regulatory Analysis

Based on the data analyses of circRNA, lncRNA, miRNA, and mRNA of goat submandibular glands, core RNAs of each group were predicted. The expression of miR-141-x was significantly negative correlated with *ITGB2*, *LCP2*, *PTPRC*, *SYK,* and *ZAP70* (r < −0.9), which indicates that the miR-141-x may play an important role in regulating the expression of core genes. The expression of *PTPRC* and *ZAP70* was significantly and highly negatively correlated with the expression of miR-141-x, miR-29-y, and chi-miR-29b-3p. 

CD45, encoded by *PTPRC*, is a highly conserved protein tyrosine phosphatase receptor, which is of great significance in the development and maturation of lymphocytes, functional regulation, and signal transduction, and plays an important role in promoting the activation and development of T cells [62]. CD45 deficiency causes T-and B-lymphocyte dysfunction, which manifests as severe combined immunodeficiency [63,64]. ZAP70, a member of the Syk family, is mainly involved in the initiation of T-cell receptor (TCR) signaling and subsequent T-cell activation and is a major tyrosine kinase protein [65]. The regulation of its expression directly affected T cell activity, which further affected the activity of goat SMGs. The regulatory relationships among core RNAs in this study imply the complexity of the regulation of gene expression by non-coding RNAs.

The analyses of four RNA molecules in the goat SMGs indicated the targeting relationships of all core circRNAs. It was predicted that novel_circ_012441 could regulate the expression of *ARHGAP35* in goat SMGs by sponge adsorption of chi-miR-130b-3p, chi-miR-15b-5p, and chi-miR-16b-5p. Similarly, novel_circ_013769 regulated the expression of *ARHGAP5* by adsorbing chi-miR-130b-3p and miR-29-y. We predicted that regulating the expression of pathogenic genes in goat SMGs at different developmental stages might inhibit the occurrence of diseases.

## 5. Conclusions

In summary, this was the first report of lncRNAs, circRNAs, and ceRNA network regulation in goat SMGs. Transcriptome analysis revealed the differentially expressed lncRNA and circRNA in salivary gland tissues of 1-month-old, 12-month-old, and 24-month-old goats. Functional enrichment and ceRNA network analysis reveals that 13 key circRNAs and 14 key lncRNAs may regulate the 5 key immune related genes ITGB2, LCP2, PTPRC, SYK, and ZAP70 through competitive binding with miR-141-x, miR-29-y, and chi-miR-29b-3p, thereby affecting the natural killer cell mediated cytotoxicity pathway, the T cell receptor signaling pathway, and other immune related pathways. Interestingly, all key genes, lncRNA, and circRNA were specifically overexpressed in salivary gland tissues of one-month-old goats, indicating that the function of key non-coding RNA molecules may be related to the gradual decline of salivary gland immune function with goat age. Although the regulatory relationship of the ceRNA network needs to be further verified, the miRNA-lncRNA/circRNA competition has certain implications for our future study on the expression regulation of immune genes in goat SMGs. The results of this study will enrich the knowledge of changes in the expression of non-coding RNAs during salivary gland development in goats and provides new insights into the relationship between non-coding RNAs and salivary gland immune function in goats.

## Figures and Tables

**Figure 1 genes-14-00187-f001:**
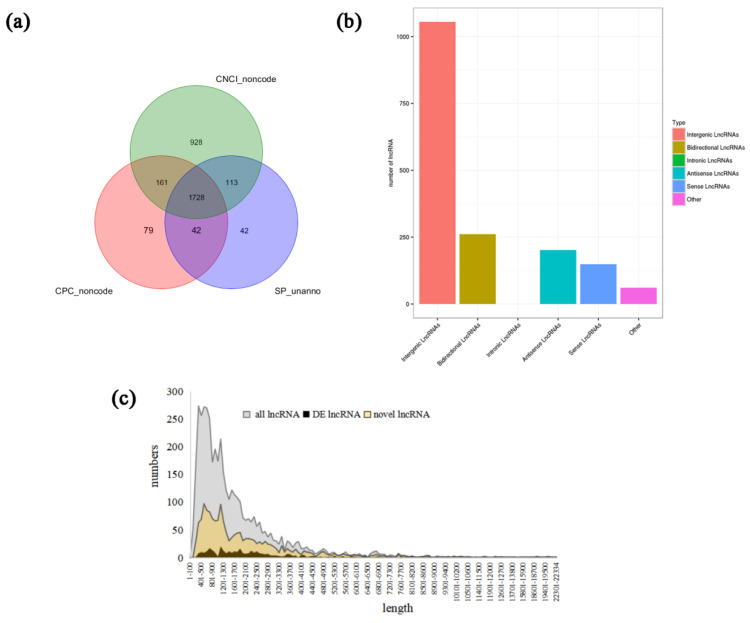
Prediction results for new lncRNAs and the statistics of new lncRNA types and lengths. (**a**) Prediction results of new lncRNAs, (**b**) Statistics of lncRNA transcript types, and (**c**) Length statistics of lncRNAs.

**Figure 2 genes-14-00187-f002:**
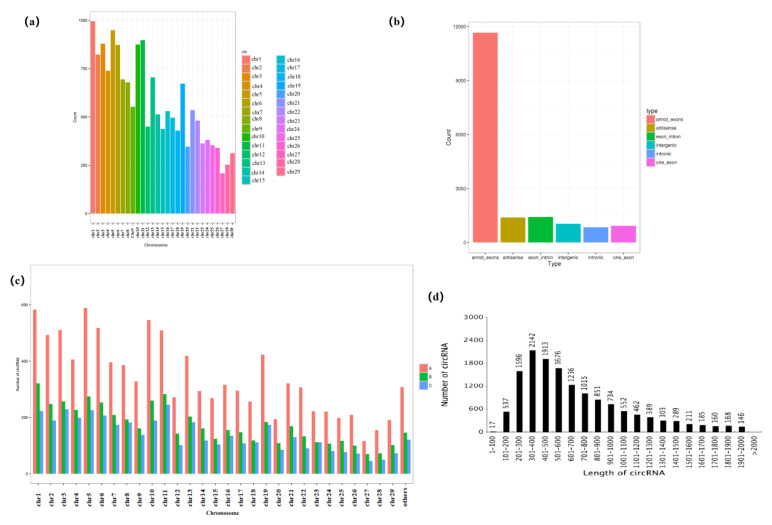
Statistics of circRNA types. (**a**) CircRNA chromosome number distribution, where the x-coordinate represents chromosomes (different chromosomes are identified by different colors) and the y-coordinate represents the number of circRNA. (**b**) CircRNA type distribution. The x-coordinate represents the type, and the y-coordinate represents the quantity. Different types of circRNA are represented in different colors: orange for annotated exons, yellow for antisense, green for exon-intron, blue-green for intergenic, blue for intronic, and red for one-exon. (**c**,**d**) The distribution diagram of the number of circRNA in each sample and circRNA length distribution.

**Figure 3 genes-14-00187-f003:**
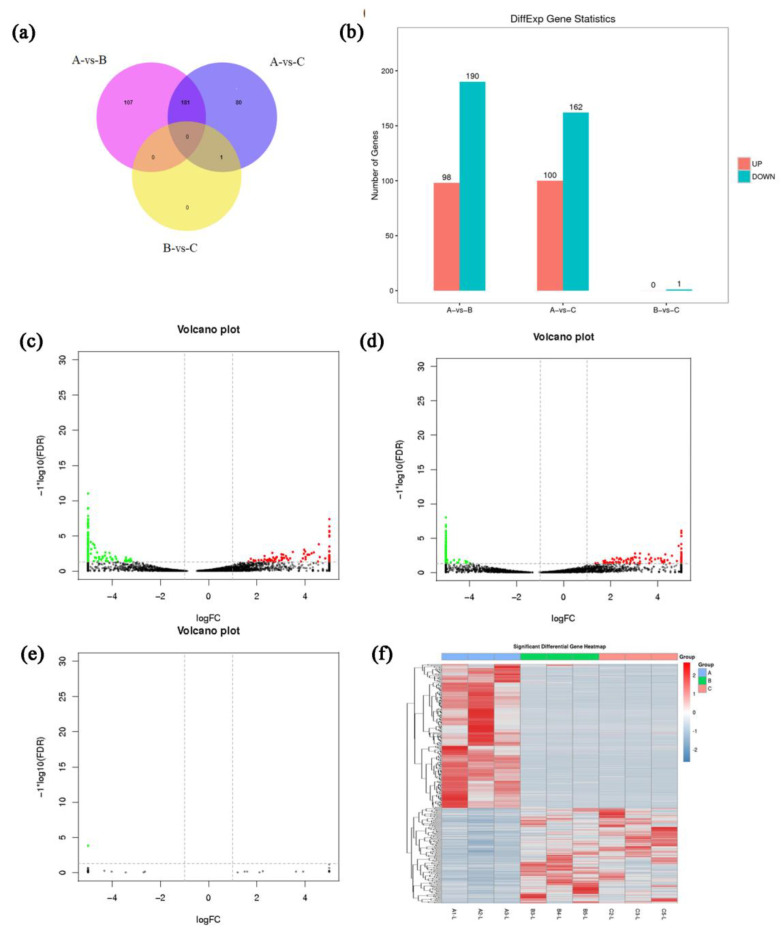
DElncRNA analysis. (**a**) Venn diagram of DElncRNAs in submandibular glands of goats at different ages. (**b**) Histogram of DElncRNAs in submandibular glands of goats at different ages. (**c**–**e**) Volcano plots of DEcircRNAs in submandibular glands of goats at different ages, red and green color represent up-regulated an down-regulated genes, respectively. (**f**) Cluster analysis of lncRNA expression patterns in submandibular glands of goats at different ages.

**Figure 4 genes-14-00187-f004:**
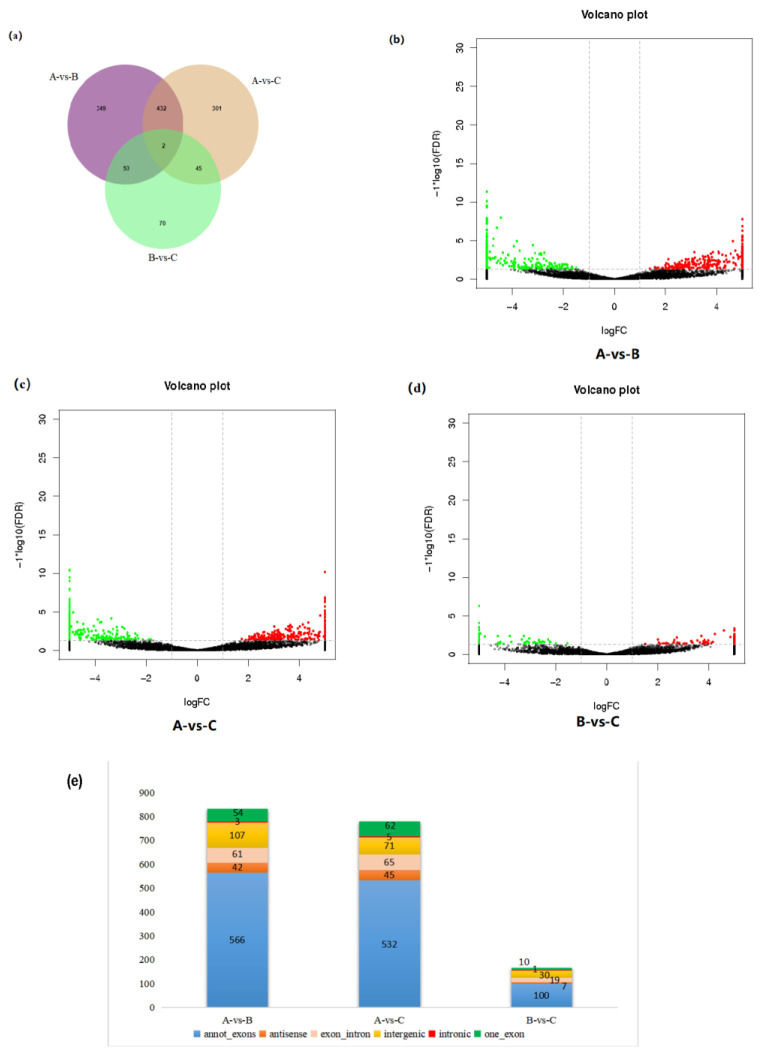
DEcircRNA analysis. (**a**) Venn diagram of DEcircRNAs in submandibular glands of goats at different ages. (**b**–**d**) Volcano plots of DEcircRNAs in submandibular glands of goats at different ages. (**e**) Bar graph and types of DEcircRNAs in submandibular glands of goats at different ages.

**Figure 5 genes-14-00187-f005:**
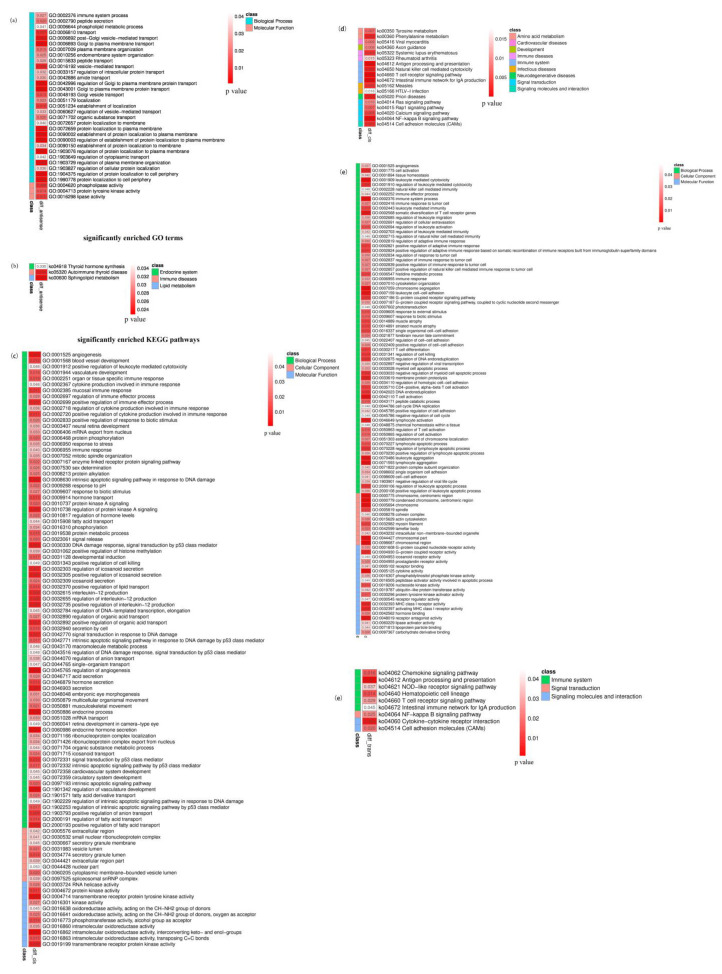
GO and KEGG pathway enrichment analyses of lncRNA targeted mRNAs with different interaction modes. (**a**,**b**) GO and KEGG pathway enrichment analyses of antisense lncRNA target mRNAs. (**c**,**d**) GO and KEGG pathway enrichment analyses of lncRNA cis-regulated target mRNAs. (**e**,**f**) GO and KEGG pathway enrichment analyses of lncRNAs that trans-regulate target mRNAs.

**Figure 6 genes-14-00187-f006:**
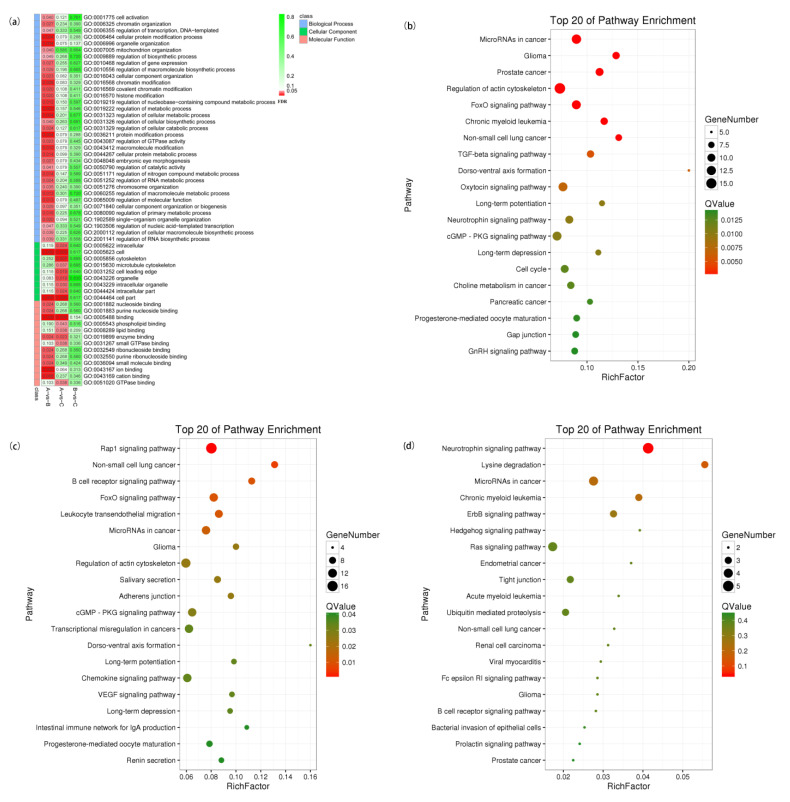
GO and KEGG enrichment analyses of DEcircRNA-hosting genes between different groups.(**a**) GO. enrichment analysis of DEcircRNA-hosting genes. (**b**) KEGG enrichment analysis of DEcircRNA-hosting genes between group A and B. (**c**) KEGG enrichment analysis of DEcircRNA hosting genes between group A and C. (**d**) KEGG enrichment analysis of DEcircRNA hosting genes between group B and C.

**Figure 7 genes-14-00187-f007:**
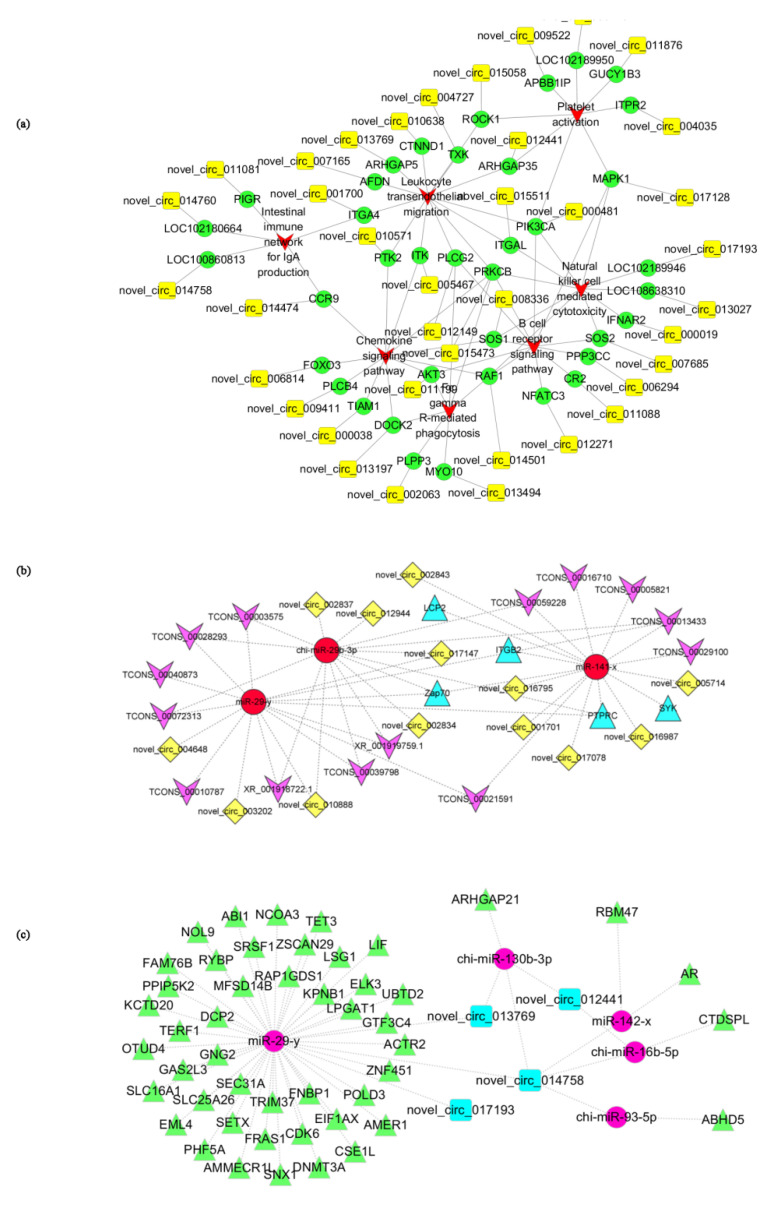
Network analysis. (**a**) Network analysis of immune-related pathways. The red V represents KEGG pathway, the green circle represents gene, and the yellow square represents circRNA. (**b**) Regulatory network of the ceRNA-miRNA-core gene. Red circles represent miRNA, rose arrows represent lncRNA, blue triangles represent core genes, and yellow rhombuses represent circRNAs. (**c**) Immune-related circRNA-miRNA-mRNA network. Red circles represent miRNA, blue squares represent circRNA, and green triangles represent mRNA.

**Figure 8 genes-14-00187-f008:**
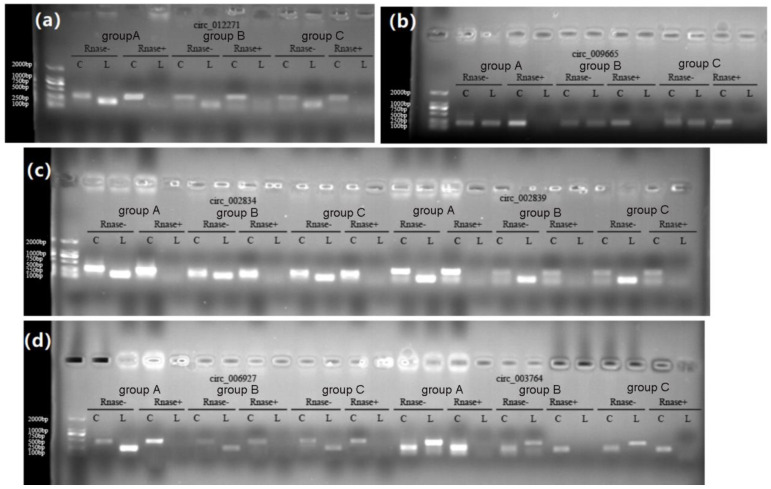
Electrophoresis of circRNA sequences at the loop junction site. (**a**–**d**) Electrophoresis of circ_012271, circ_009665, circ_002834, circ_002839, circ_006927, circ_003764 sequences at the loop junction site, respectively.

**Figure 9 genes-14-00187-f009:**
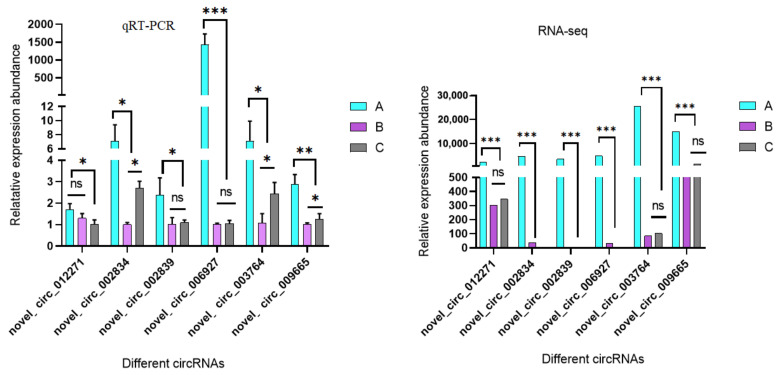
qRT-PCR validation of the differential expressed circRNAs. * indicates *p* < 0.05, ** indicates *p* < 0.01, *** indicates *p* < 0.001, ns indicates there is no significant differences.

**Table 1 genes-14-00187-t001:** Primer sequences of DEcircRNAs for qRT-PCR.

Name	Primer Sequences for Circle Amplification		Primer Sequences for Line Amplification	
novel_circ_012271	F R	AAGGTAGCCGAGGAGCAGTA	F	ACCTCCACTAGACTGGCCTT
		GGGACCACTTAATGGGCTGT	R	GGCCGTAAGTATCGGTCGTC
novel_circ_002834	F	CCTATGAGCAGTATTTCGGCCC	F	CCCTGAGCCCTACAAAGAGG
	R	GGAGTCATTCTGCGTGAGGT	R	AGCCCGTAGAACTGGACTTG
novel_circ_002839	F	ACCCAGTACTTTGGGCCTG	F	CCCTGAGCCCTACAAAGAGG
	R	ACTGGACTTGGCATCGGAAG	R	AGCCCGTAGAACTGGACTTG
novel_circ_006927	F	CCCTCCGAAACATCCCTCAG	F	GGCACGTTGAAGGTTGTGTG
	R	AGGTCATAGGGGGTTGGGAA	R	TGAGCATGATTCGCTCACCT
novel_circ_003764	F	GAGAAGTGGTGGCTTCCAGAC	F	GAGAAGTGGTGGCTTCCAGAC
	R	TTTCGGGCAGTGGTAAAGGC	R	GCCATGCCCATTCTTTGCTT
novel_circ_009665	F	TTTTCCGTGCTCTTGGAGGT	F	TTTTCCGTGCTCTTGGAGGT
	R	GCCGTGGGTTTTCTTTTGCC	R	GGCAACACTGTCTGGTCTGA
novel_circ_017193	F	CCCACCTATGAGTTGCTGGT	F	TTGGTGCTGTCAATAAAGCG
	R	GTCCCGGGGGTTGCATATTT	R	GGCTCACCAGATTCCCACTC

**Table 2 genes-14-00187-t002:** Statistics of differentially expressed lncRNAs in submandibular glands of goats at different ages.

Pairs	Diffgenes (Up)	Diffgenes (Down)	All Diffgenes
A vs. B	98	190	288
A vs. C	100	162	262
B vs. C	0	1	1

**Table 3 genes-14-00187-t003:** Statistics of DEcircRNAs in submandibular glands of goats at different ages.

Pairs	Up-Regulated circRNAs	Down-Regulated circRNAs	All DEcircRNAs
A vs. B	355	478	833
A vs. C	349	431	780
B vs. C	86	81	167

**Table 4 genes-14-00187-t004:** Regulation-type statistics of DElncRNAs and DEGs.

Type	Number of DElncRNAs	Targeted Gene Number
Antisense analysis	11	10
Cis-regulation	51	70
Trans-regulation	160	141

**Table 5 genes-14-00187-t005:** Functional enrichment analysis results of part antisense lncRNA target mRNAs.

lncRNA	mRNA	Symbol	KEGG Pathway	GO Term
XR_310691.3 XR_001919471.1 XR_001919469.1	XM_005692177.3	*SMPD3*	Sphingolipid metabolism	phospholipase activity
				lipase activity
				peptide secretion
				immune system process
				peptide transport
				phospholipid metabolic process
TCONS_00062254	XM_018066366.1	*ncbi_102170270*	/	protein tyrosine kinase activity
				immune system process
TCONS_00044048	XM_018058585.1	*ncbi_100750237*	Autoimmune thyroid disease	/
			Thyroid hormone synthesis	

**Table 6 genes-14-00187-t006:** Analysis of lncRNAs and cis-regulated target mRNAs in the first four significantly enriched pathways.

lncRNA_ID	chr	Strand	Target Gene ID	Symbol	Up/Down_Stream	Distance	r	*p*_Value
TCONS_00011775	NC_030811.1	+	TCONS_00013166	/	DOWNSTREAM	2254	0.33	3.92 × 10^−1^
** *TCONS_00013148* **	** *NC_030811.1* **	*-*	** *TCONS_00013145* **	** *V-TCR* **	** *UPSTREAM* **	** *1145* **	** *0.93* **	**2.46 × 10^−4^**
** *TCONS_00013148* **	** *NC_030811.1* **	*-*	** *TCONS_00013152* **	** *V-TCR* **	** *DOWNSTREAM* **	** *4025* **	** *0.95* **	**7.74 × 10^−5^**
** *TCONS_00013148* **	** *NC_030811.1* **	*-*	** *XM_013963475.2* **	** *FYN* **	** *UPSTREAM* **	** *17,393* **	** *0.99* **	**2.26 × 10^−7^**
TCONS_00028591	NC_030816.1	+	XM_018053054.1	FYN	DOWNSTREAM	3690	0.61	8.36 × 10^−2^
** *TCONS_00028592* **	** *NC_030816.1* **	** *+* **	** *XM_018053054.1* **	** *FYN* **	** *DOWNSTREAM* **	** *4978* **	** *0.91* **	**5.61 × 10^−4^**
TCONS_00029100	NC_030816.1	+	XM_018053345.1	LOC102177708	DOWNSTREAM	25,010	0.86	2.82 × 10^−3^
** *TCONS_00029100* **	** *NC_030816.1* **	** *+* **	** *XM_018053341.1* **	** *LOC102185264* **	** *UPSTREAM* **	** *2709* **	** *0.93* **	**2.43 × 10^−4^**
TCONS_00068293	NC_030830.1	-	XM_005696727.3	LOC102178027	UPSTREAM	20,238	0.27	4.89 × 10^−1^
TCONS_00068293	NC_030830.1	-	XM_018038547.1	LOC102178318	UPSTREAM	20,964	0.20	6.05 × 10^−1^
TCONS_00068294	NC_030830.1	-	XM_005696727.3	LOC102178027	UPSTREAM	21,924	0.66	5.10 × 10^−2^
TCONS_00068294	NC_030830.1	-	XM_018038547.1	LOC102178318	UPSTREAM	22,650	0.63	7.01 × 10^−2^
TCONS_00080824	NW_017189517.1	+	XM_005700359.3	SH2D1A	DOWNSTREAM	11,734	0.83	5.44 × 10^−3^
XR_001917128.1	NC_030830.1	-	XM_005696604.3	LTB	DOWNSTREAM	7661	0.54	1.37 × 10^−1^

**Table 7 genes-14-00187-t007:** Statistics of lncRNA trans-regulated genes in significantly enriched immune-related pathways.

lncRNA_id	mRNA_id	Symbol	Type	Pearson Cor	*p*-Value
TCONS_00041162	TCONS_00013152	*V-TCR*	pos	0.9999	3.75 × 10^−14^
TCONS_00052910	TCONS_00013152	*V-TCR*	pos	0.9999	1.51 × 10^−14^
XR_001297267.2	TCONS_00013169	*TCRB*	pos	0.9998	1.70 × 10^−13^
TCONS_00072725	TCONS_00040879	*PRKCQ*	pos	0.9996	5.92 × 10^−12^
TCONS_00078593	TCONS_00063963	*HSP90AA1*	pos	0.9821	2.48 × 10^−6^
TCONS_00011740	XM_005677288.3	*LOC102179433*	pos	0.9999	5.77 × 10^−15^
XR_001919673.1	XM_005690341.3	*MS4A1*	pos	0.9992	5.05 × 10^−11^
XR_310184.3	XM_005692018.3	*CCL22*	pos	0.9999	7.11 × 10^−15^
TCONS_00007637	XM_005692019.3	*LOC102169211*	pos	1.0000	2.66 × 10^−15^
TCONS_00020715	XM_005693201.3	*CCL5*	neg	−0.9648	2.60 × 10^−5^
TCONS_00035813	XM_005694804.3	*IL7R*	pos	0.9980	1.15 × 10^−9^
TCONS_00072236	XM_005695522.3	*CCR4*	pos	0.9996	5.66 × 10^−12^
TCONS_00013889	XM_018038885.1	*LOC102181347*	pos	0.9978	1.69 × 10^−9^

**Table 8 genes-14-00187-t008:** Immune-related circRNA information.

circRNA ID	Source_Gene ID	Source_Gene Symbol	Annot_Type	High Expression
novel_circ_012271	ncbi_102180958	NFATC3	annot_exons	A
novel_circ_017193	ncbi_102189946	LOC102189946	exon_intron	A
novel_circ_011088	ncbi_102186615	CR2	annot_exons	A
novel_circ_004727	ncbi_102188026	TXK	annot_exons	A
novel_circ_014760	ncbi_102180664	LOC102180664	annot_exons	A
novel_circ_012149	ncbi_102180337	PLCG2	annot_exons	A
novel_circ_013769	ncbi_102178574	ARHGAP5	one_exon	B
novel_circ_012441	ncbi_102177274	ARHGAP35	one_exon	B
novel_circ_011081	ncbi_102176714	PIGR	antisense	B
novel_circ_014758	ncbi_100860813	LOC100860813	exon_intron	C

**Table 9 genes-14-00187-t009:** The negative correlation expressed between miRNAs and core genes.

miRNA Gene		*CCR7*	*CD28*	*ITGB2*	*LCP2*	*MYC*	*PTPRC*	*SELL*	*SYK*	*ZAP70*
chi-miR-193b-3p	r	−0.743 *	−0.778 *	−0.769 *	−0.782 *	−0.732 *	−0.796 *	/	−0.800 **	−0.756 *
	*p*	0.022	0.014	0.015	0.013	0.025	0.01	/	0.01	0.018
chi-miR-29b-3p	r	−0.774 *	−0.853 **	−0.856 **	−0.859 **	−0.796*	−0.919 **	−0.795 *	−0.884 **	−0.946 **
	*p*	0.014	0.003	0.003	0.003	0.01	0	0.01	0.002	0
miR-141-x	r	−0.832 **	−0.884 **	−0.909 **	−0.900 **	−0.836 **	−0.949 **	−0.824 **	−0.928 **	−0.928 **
	*p*	0.005	0.002	0.001	0.001	0.005	0	0.006	0	0
miR-148-x	r	/	/	/	/	/	−0.695 *	/	−0.686 *	−0.698 *
	*p*	/	/	/	/	/	0.038	/	0.041	0.037
miR-148-y	r	−0.699 *	−0.732 *	−0.747 *	−0.745 *	−0.679 *	−0.771 *	/	−0.773 *	−0.733 *
	*p*	0.036	0.025	0.021	0.021	0.044	0.015	/	0.015	0.025
miR-29-y	r	−0.766 *	−0.848 **	−0.859 **	−0.855 **	−0.775 *	−0.921 **	−0.812 **	−0.891 **	−0.954 **
	*p*	0.016	0.004	0.003	0.003	0.014	0	0.008	0.001	0
miR-6516-x	r	/	−0.711 *	−0.731 *	−0.726 *	−0.680 *	−0.780 *	−0.703 *	−0.758 *	−0.794 *
	*p*	/	0.032	0.025	0.027	0.044	0.013	0.035	0.018	0.011
miR-885-y	r	−0.671 *	−0.696 *	−0.720 *	−0.711 *	−0.688 *	−0.739 *	/	−0.721 *	−0.706 *
	*p*	0.048	0.037	0.029	0.032	0.04	0.023	/	0.028	0.033
novel-m0357-5p	r	/	−0.672 *	−0.680 *	−0.679 *	-0.633	−0.733 *	/	−0.705 *	−0.760 *
	*p*	/	0.047	0.044	0.044	0.067	0.025	/	0.034	0.018

** represents a significant correlation at level 0.01 (double-tailed), * represents a significant correlation at level 0.01 (double-tailed).

## Data Availability

The original contributions presented in the study are publicly available. All data can be downloaded from the Gene Expression Omnibus (GEO) database, and the accession number is: GSE144368.

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
