# Peer review of "Analyses of lncRNAs, circRNAs, and the Interactions between ncRNAs and mRNAs in Goat Submandibular Glands Reveal Their Potential Function in Immune Regulation"

_genes, 2023, doi:10.3390/genes14010187_

Round 1

Reviewer 1 Report

The authors have one a great effort to conduct such a study and compile this manuscript. 

Below mentioned are some of my comments:

1. The Conclusion may be improvised:

I noticed a brief paragraph just before the conclusion section which represent a conclusion and future perspective for the study, hence this may bring in a small repetition. Furthermore, the actual conclusion section lacks quality information. Hence I suggest that you look into this: avoid immediate repetition of conclusion matter twice and/or shorten the conclusion paragraph in the discussion and elaborate the portion under the subsection 'conclusion'

2. Abstract/Conclusion improvement: Also, it would be even better if you could bring in some vital findings of your study (key lncRNA or circRNA or ceRNA or gene or even some functional pathway) that demarcates the studied animal group and mention them either in the abstract or conclusion

3. How would the authors address the possibility of difference in expression due to difference in submandibular gland activity due to changes in feeding pattern of the goats? The 1 month-old goat kids would be feed only on the dam's milk/replacer milk while the 12 and 24 month would have access to roughage and concentrate (that is the usual practice which I suppose would have been followed here too) so wouldn't this affect the gland's activity and thereby the expression profile?

Author Response

Point 1. The Conclusion may be improvised:

I noticed a brief paragraph just before the conclusion section which represent a conclusion and future perspective for the study, hence this may bring in a small repetition. Furthermore, the actual conclusion section lacks quality information. Hence I suggest that you look into this: avoid immediate repetition of conclusion matter twice and/or shorten the conclusion paragraph in the discussion and elaborate the portion under the subsection 'conclusion'.

Response 1: Thank you for your comment! We have deleted the conclusion paragraph in the discussion and rewritten our actual conclusion section as follows:

“In summary, this was the first report of lncRNAs, circRNAs, and ceRNA network regulation in goat SMGs. Transcriptome analysis revealed the differentially expressed lncRNA and circRNA in salivary gland tissues of 1-month-old, 12-month-old and 24-month-old goats. Functional enrichment and ceRNA network analysis reveals that 13 key circRNAs and 14 key lncRNAs may regulate the 5 key immune related genes ITGB2, LCP2, PTPRC, SYK and ZAP70 through competitive binding with miR-141-x, miR-29-y and chi-miR-29b-3p, thereby affecting the Natural killer cell mediated cytotoxicity pathway, T cell receptor signaling pathway, and other immune related pathways. Interestingly, all key genes, lncRNA, and circRNA were specifically overexpressed in salivary gland tissues of one-month-old goats, indicating that the function of key non-coding RNA molecules may be related to the gradual decline of salivary gland immune function with goat age. Although the regulatory relationship of the ceRNA network needs to be further verified, the miRNA-lncRNA/circRNA competition has certain implications for our future study on the expression regulation of immune genes in goat SMGs. The results of this study will enrich the knowledge of changes in the expression of non-coding RNAs during salivary gland development in goats, and provides new insights into the relationship between non-coding RNAs and salivary gland immune function in goats.”(Line 711-728)

Point 2. Abstract/Conclusion improvement: Also, it would be even better if you could bring in some vital findings of your study (key lncRNA or circRNA or ceRNA or gene or even some functional pathway) that demarcates the studied animal group and mention them either in the abstract or conclusion

Response 2: We fully agree with your comment! We revised the contents of Abstract and Conclusion, summarized the key ncRNAs and genes found, and pointed out their expression characteristics in the salivary glands of goats at different ages. The revised Abstract is as follows:

“Abstract: As one of the main ruminants, goat salivary glands hardly secrete digestive enzymes but play an important role in immunity. The immune function of goat salivary glands significantly changes with age, while the expression profile and specific function of non-coding RNA during this process are unknown. In this study, transcriptome sequencing was performed on salivary gland tissues of 1-month-old, 12-month-old, and 24-month-old goats, revealing the expression patterns of lncRNA and circRNA at different ages. A total of 369 lncRNAs and 1699 circRNAs were found to be differentially expressed. Functional enrichment analyses showed that the lncRNA regulated target mRNAs and circRNA host genes were significantly enriched in immune-related GO terms and pathways. CeRNA network analysis showed that the key differentially expressed circRNAs and lncRNAs mainly regulate the key immune-related genes ITGB2, LCP2, PTPRC, SYK, and ZAP70 through competitive binding with miR-141-x, miR-29-y, and chi-miR-29b-3p, thereby affecting the natural killer cell-mediated cytotoxicity pathway, the T cell receptor signaling pathway, and other immune-related pathways. It should be noted that the expression of key circRNAs, lncRNAs, and key immune-related genes in goat SMGs decreased significantly with the growth of the goat. This is the first reporting of lncRNAs, circRNAs, and ceRNA network regulation in goat SMGs. Our study contributes to the knowledge of changes in the expression of non-coding RNAs during salivary gland development in goats, and provides new insights into the relationship between non-coding RNAs and salivary gland immune function in goats.” (Line 14-32)

Point 3. How would the authors address the possibility of difference in expression due to difference in submandibular gland activity due to changes in feeding pattern of the goats? The 1 month-old goat kids would be feed only on the dam's milk/replacer milk while the 12 and 24 month would have access to roughage and concentrate (that is the usual practice which I suppose would have been followed here too) so wouldn't this affect the gland's activity and thereby the expression profile?

Response 3: Thank you for your kindly reminding! In the process of experimental design, we did worry about whether the change of feed composition would cause batch effect, but we finally decided to carry out the research according to the current method. The main reason is that we are worried that if we adopt early weaning and other methods to unify the diet of goats at different ages, the salivary gland function of 1-month old goats may change to that of adult goats in advance, which is not conducive to our research on the differences in salivary gland turnover between goat kids and adult goats.

Reviewer 2 Report

The present manuscript describes the regulation of lncRNAS, circRNAS, and ceRNA network in goat SMGs during the salivary gland development which influences the immune regulation. The authors have analysed and interpreted the data scientifically. The results are quite convincing.  However, the use of reference gene selection for the validation of circRNA needs to be revisited. In goats, 18S, TBP, and HMBS are preferred reference gene for analyzing gene expression studies (Zhang et al.,2013). In the present study, GAPDH has been used, which is the stable reference gene for analyzing PBMCs rather than the tissues. Thus, if it could be replaced with anyone /combinations of aforementioned reference genes, the results will be more accurate. 

Author Response

Point: The present manuscript describes the regulation of lncRNAS, circRNAS, and ceRNA network in goat SMGs during the salivary gland development which influences the immune regulation. The authors have analysed and interpreted the data scientifically. The results are quite convincing.  However, the use of reference gene selection for the validation of circRNA needs to be revisited. In goats, 18S, TBP, and HMBS are preferred reference gene for analyzing gene expression studies (Zhang et al.,2013). In the present study, GAPDH has been used, which is the stable reference gene for analyzing PBMCs rather than the tissues. Thus, if it could be replaced with anyone /combinations of aforementioned reference genes, the results will be more accurate.

Response: Thank you for your very professional advice! During the preparation of the qRT-PCR experiment, we also considered searching for appropriate reference genes through a literature review. As there is no research report on qRT-PCR reference genes in goat salivary gland tissues, we have referred to the selection of internal reference genes in salivary gland qRT-PCR tests in other species. For example, GAPDH is used as the only reference gene in mouse submandibular gland tissues (Gong et al., 2021), human salivary gland epithelial cells (Noll et al., 2022), and human salivary gland tissue (Aqrawi et al., 2020).

Of course, if you think it is really necessary to use 18S, TBP, and HMBS as reference genes, we are willing to re-implement the qRT-PCR test.

References:

Gong, W., Qiao, Y., Li, B., Zheng, X., Xu, R., Wang, M., Mi, X., & Li, Y. (2021). The Alteration of Salivary Immunoglobulin A in Autism Spectrum Disorders. Frontiers in psychiatry, 12, 669193.

Noll, B., Mougeot, F. B., Brennan, M. T., & Mougeot, J. C. (2022). Regulation of MMP9 transcription by ETS1 in immortalized salivary gland epithelial cells of patients with salivary hypofunction and primary Sjögren's syndrome. Scientific reports, 12(1), 14552.

Aqrawi, L. A., Jensen, J. L., Fromreide, S., Galtung, H. K., & Skarstein, K. (2020). Expression of NGAL-specific cells and mRNA levels correlate with inflammation in the salivary gland, and its overexpression in the saliva, of patients with primary Sjögren's syndrome. Autoimmunity, 53(6), 333–343.
